# Updates in Diagnosis and Endoscopic Management of Cholangiocarcinoma

**DOI:** 10.3390/diagnostics14050490

**Published:** 2024-02-24

**Authors:** Roxana-Luiza Caragut, Madalina Ilie, Teodor Cabel, Deniz Günșahin, Afrodita Panaitescu, Christopher Pavel, Oana Mihaela Plotogea, Ecaterina Mihaela Rînja, Gabriel Constantinescu, Vasile Sandru

**Affiliations:** 1Clinical Department of Gastroenterology, Bucharest Emergency Clinical Hospital, 014461 Bucharest, Romania; roxanacaragut@gmail.com (R.-L.C.); drsandruvasile@gmail.com (V.S.); 2Department of Gastroenterology, University of Medicine and Pharmacy “Carol Davila” Bucharest, 050474 Bucharest, Romania

**Keywords:** cholangiocarcinoma, diagnosis, endoscopy, biliary drainage, digestive oncology, endoscopic retrograde cholangiopancreatography, endoscopic ultrasound

## Abstract

Cholangiocarcinoma (CCA) is an adenocarcinoma originating from the epithelial cells of the bile ducts/hepatocytes or peribiliary glands. There are three types of cholangiocarcinoma: intrahepatic, perihilar and distal. CCA represents approximately 3% of the gastrointestinal malignancies. The incidence of CCA is higher in regions of the Eastern world compared to the Western countries. There are multiple risk factors associated with cholangiocarcinoma such as liver fluke, primary sclerosing cholangitis, chronic hepatitis B, liver cirrhosis and non-alcoholic fatty liver disease. Endoscopy plays an important role in the diagnosis and management of cholangiocarcinoma. The main endoscopic methods used for diagnosis, biliary drainage and delivering intrabiliary local therapies are endoscopic retrograde cholangiopancreatography and endoscopic ultrasound. The purpose of this review is to analyze the current data found in literature about cholangiocarcinoma, with a focus on the actual diagnostic tools and endoscopic management options.

## 1. Introduction

Cholangiocarcinoma (CCA) is an adenocarcinoma (rarely adenosquamous carcinoma/clear cell carcinoma) that has its origin in the epithelial cells of the bile ducts, but it can also arise from hepatocytes or peribiliary glands. CCA can appear in any segment of the biliary tract or in the hepatic parenchyma [1]. Anatomically, CCA can be classified as intrahepatic (iCCA), perihilar (pCCA) and distal (dCCA). Intrahepatic CCA is localized in the liver parenchyma, proximal to the second-order bile ducts; perihilar CCA develops between the second-order bile ducts and the fusion of cystic duct with the common bile duct; and distal CCA is limited to the common bile duct below the cystic duct insertion [2]. Each of these types is characterized by different clinical presentations, genetic aberrations and management [3]. 

## 2. Epidemiology

CCA represents approximately 3% of the gastrointestinal malignancies. The incidence of CCA is higher in regions of the Eastern world (in countries where liver fluke is endemic) compared to Western countries. Even between regions of the same country there is significant difference. The highest incidence is found in Thailand, with an age-standardized incidence rate/100,000 individuals of 85 in the Northeast region, 14.5 for the North and Central region and 5.7 for the South region. In South Korea, the incidence is 8.8 per 100,000 individuals in Gwangju and 7.1 in Busan. The lowest incidence of CCA is found in Israel and Costa Rica—0.3 per 100,000 individuals [4,5]. CCA affects mainly men and older individuals [6]. In Europe, the incidence ranges between 0.4/100,000 and 1.8/100,000 individuals and in USA from 0.6/100,000 to 1.0/100,000 individuals [7].

## 3. Risk Factors

There are a series of risk factors associated with CCA development. Opisthorchis viverrini is a trematode that is classified as a group 1 biological carcinogen. It causes liver fluke infection and it is a major cause of CCA in Southeast Asia. Other parasites that are associated with CCA are Clonorchis sinensis (clonorchiasis) and Opisthorchis felineus. The consumption of undercooked cyprinid fish (carrying the larval parasite) can cause the infection [8,9]. Liver flukes attach to the biliary wall leading to ulceration and precancerous lesions. Moreover, those parasites promote chronic inflammation by increasing the secretion of inflammatory cytokines like IL-6 (an important link between inflammation and carcinogenesis). In O. viverrini infection, IL-6 is involved in the development of periductal fibrosis, which is considered to be implicated in CCA development [1]. In Western countries, the primary sclerosing cholangitis (PSC) is the most important risk factor for CCA [4]. CCA is the most common neoplasia in patients with primary sclerosing cholangitis (PSC) and has a high rate of mortality. CCA’s pathogenesis in patients with PSC is unknown but the underlying factors are represented by chronic inflammation and genetic or epigenetic abnormalities. In the first year succeeding the diagnosis of PSC, 30–50% of patients are diagnosed with CCA. The annual incidence varies between 0.5 and 1.5 per 100 persons, which is approximately 10 to 1000 times greater than the incidence in the general population. Usually, CCA in patients with PSC develops from a dominant stricture. A dominant stricture is a stricture with a diameter less than 1 mm localized in the hepatic duct or less than 1.5 mm localized in the common bile duct [10].

Intrahepatic CCA is more associated with chronic hepatitis B, liver cirrhosis and non-alcoholic fatty liver disease (NAFLD). Choledocolithiasis is strongly associated with perihilar CCA and distal CCA. Choledochal cysts and Caroli disease, type 2 diabetes mellitus, obesity and cigarette smoking are associated with all of the three types of CCA. Chronic pancreatitis is associated with distal CCA [11].

## 4. Genetic Factors

The molecular alterations of cholangiocarcinoma differ appreciably by etiology and subtypes. For example, FGFR2 fusions are found only in iCCA with a prevalence of 10–15%. PRKCA–PRKCB fusions are found in dCCA and pCCA [12,13]. There are a series of genetic mutation associated with CCA. In fluke-related CCA the most frequent mutation are represented by ACVR1B, ARID1A, FBXW7, MAP2K4, MSH3, SMAD4, BRCA1, H3K27me3-associated promoter mutations, PTEN and TP53 and also ERBB2 amplification with a major clinical implication due to the tumor’s response to ERBB2 inhibitor treatment. In non-fluke-related CCA, IDH1 and 2 (gain-of-function mutations) as well as BAP1 can be present. There are several mutations that are associated with both fluke and non-fluke-related CCA, for example APC, BRAF, BRCA1 and 2, CDKN1B, CTNNB1, ELF3, KRAS, P4HTM, RASA1, RB1, STK11, TGFBR2, ACVR2A, ARID2, NCOR1, NRAS, PBRM1, PIK3R1, RNF43, SF3B1 and ASXL1. Epigenetically, CCA tumors show ADN hypermethylation, more precisely CpG island hypermethylation in fluke-related CCA and CpG shore hypermethylation in non fluke-related CCA. This aspect is helpful in differentiating the two types [14].

## 5. Macroscopic and Microscopic Features of CCA

Regarding the growth patterns, macroscopically CCA can present as mass-forming/periductal infiltrating/intraductal papillary lesions. The first two patterns have the poorest prognosis, meanwhile the intraductal papillary lesions have a good prognosis after curative surgical resection. Intrahepatic CCA presents mostly as a mass-forming lesion. Perihilar CCA presents usually as a periductal infiltrating lesion which is developing along the walls of large bile ducts leading to strictures and spreading through the portal tracts. The intraductal papillary lesions present as a papillary or polypoid tumor that develops within the lumen of a bile duct [15,16].

Microscopically, the majority of cholangiocarcinomas are well, moderately or poorly differentiated adenocarcinomas (90–95%). Intrahepatic CCA is an adenocarcinoma respecting the above-mentioned degrees of differentiation. According to the AJCC/UICC (American Joint Committee on Cancer/Union for International Cancer Control) and WHO (World Health Organization) classification systems, ICCA can derive from large or small bile ducts. The large bile duct types contain taller columnar cells that form larger tubules and show mucin production and the small bile duct types contain cuboidal cells, forming small tubular/trabecular structures and show no or only minimal mucin production [17]. PCCa and dCCA have similar characteristics as iCCA deriving from large bile ducts [18].

Cholangiocarcinoma can also present as a rare variant such as combined hepatocellular carcinoma and CCA, intestinal-type CCA, cholangiolocellular carcinoma that contains cholangioles and it is considered to be a histological subtype of well-differentiated intrahepatic CCA [19,20], squamous/adenosquamous CCA, lymphoepithelial CCA, mucinous/signet ring, clear cell CCA and undifferentiated CCA [21].

## 6. Diagnosis

### 6.1. Clinical Presentation

The signs and symptoms of cholangiocarcinoma are different depending on its location.

Patients with periductal and distal CCA can experience symptoms related to biliary obstruction such as painless jaundice, clay-colored stools, dark urine and pruritus. Other symptoms include weight loss, fatigue, night sweats, malaise and abdominal pain in the right upper quadrant. In some cases, the first clinical presentation can be cholangitis [1].

In intrahepatic CCA, symptoms are represented by dull pain in the right upper quadrant, weight loss, malaise and about 25–30% of the patients do not experience any symptoms, the tumor being detected incidentally on imaging examinations [1,22].

During physical examination, patients may exhibit jaundice, a mass in the right upper quadrant, and occasionally a palpable gallbladder (known as the Courvoisier sign), which is typically observed in perihilar cholangiocarcinoma (pCCA) and distal cholangiocarcinoma (dCCA). In iCCA, tenderness in the right upper quadrant and signs of weight loss can be detected [23].

### 6.2. Laboratory Findings

Laboratory tests usually reveal an elevated total bilirubin, direct bilirubin, alkaline phosphatase and gamma-glutamyl transpeptidase. Elevated aspartate aminotransferase (AST), alanine aminotransferase (ALT) levels, as well as elevated internationalized normalized ratio (INR) prolonged prothrombin time are found in chronic biliary obstruction as a mark of liver dysfunction [1]. In iCCA, alkaline phosphatase is elevated, and the bilirubin levels are usually normal. A part of the patients diagnosed with CCA can have altered laboratory findings suggestive for hypercalcemia of malignancy (hypercalcemia associated with hypophosphatemia, low parathyroid hormone and vitamin D levels) [22,23].

CA (carbohydrate antigen) 19-9, CA 125 and carcinoembryonic antigen (CEA) represent the most universally used biomarkers for diagnosis and mostly for monitoring cholangiocarcinoma [24]. A study proved that CA 19-9 value over 129 U/mL in patients with PSC has a sensitivity of 76% and a specificity of 98% in diagnosing CCA. The fact that CA 19-9 value can be elevated in other pathologies (for example, in cholestasis of benign causes) and that it is absent in 7% of the general population has to be taken into account. Frequently, a value of CA 19-9 over 1000 U/mL is associated with the presence of metastatic CCA [25]. Other serum biomarkers like CA 242 and cytokeratin-19 were reported to be useful in diagnosing CCA but are not yet used in clinical practice [26].

### 6.3. Imaging of CCA

Imaging plays an important role in the diagnosis, staging and follow up of CCA. Ultrasound (US) or contrast-enhanced ultrasound (CEUS), computed tomography (CT), MRI (Magnetic Resonance Imaging) and PET (positron emission tomography) are the most common imaging methods used [1].

#### 6.3.1. Ultrasound (US)

Abdominal ultrasound has a high detection rate of intrahepatic CCA. It can be seen as an intrahepatic mass associated with a distension of the upstream bile ducts [27]. Nevertheless, the differential diagnosis between HCC (hepatocellular carcinoma) and iCCA is difficult using only abdominal ultrasound, especially on a cirrhotic liver. Transabdominal ultrasound has an accuracy of around 80–95% in the identification of dCCA but pCCA is difficult to diagnose using this imaging method. Transabdominal US is often used as the first imaging method in order to exclude a benign cause of the bile duct obstruction and to guide the election of the next method in order to describe the mass [27,28]. Periductal infiltrating tumors are associated with narrowed or dilated bile ducts without an apparent mass. Intraductal tumors are associated with duct ectasia with/without a visible mass. If a mass is visible, it is usually polypoid and hyperechoic in comparison to the surrounding liver [29].

#### 6.3.2. Abdominal Contrast-Enhanced Ultrasound (CEUS)

In about 50% of the intrahepatic CCAs, at CEUS examination, the tumor has a homogeneous arterial hyperenhancement. The majority of iCCAs show peripheral rim-like hyperenhancement followed by an early washout (usually in less than 60 s after contrast injection). HCC commonly has a homogeneous or inhomogeneous (in large nodules) and intense hyperenhancement followed by washout in the late phase, usually with an onset at 2 min after injection. It is important to mention that neither US nor CEUS are used for tumor staging [30].

#### 6.3.3. Computed Tomography (CT) Imaging

CT is the standard imaging method used to characterize, stage and follow-up cholangiocarcinoma [31].

Mass-forming cholangiocarcinomas have a lower density compared to the surrounding liver on native CT scan. After contrast administration, CCA shows a peripheral (arterial peripheral-rim enhancement) with gradual centripetal enhancement [29]. The grade of centripetal enhancement depends on the degree of fibrosis existent in the center of the mass. Fibrotic areas are represented by delayed phase enhancement meanwhile the hypercellular components correspond on arterial phase to areas of hyperenhancement. Apparently, the hypercellular component has a better prognosis in terms of disease-free overall survival [29,32]. Capsular retraction, satellite nodules and dilatation of the distal bile ducts can also be seen on CT scan. Vascular invasion can also be present, and it is usually associated with lobar or segmental hepatic atrophy. CCA is usually narrowing the portal or hepatic veins but without forming a tumor thrombus, compared to HCC [33].

Periductal infiltrating tumors present on CT imaging as areas of duct wall/periductal parenchyma thickening and cause the narrowing of the involved bile duct and the dilatation of the proximal biliary tree. Compared to the benign strictures, cholangiocarcinoma is usually longer and shows contrast enhancement [34].

Intraductal tumors are represented by alterations in duct caliber (duct ectasia) with/without a visible mass on CT scan. If a mass is seen, it is usually polypoid. It is hypodense on a precontrast CT and shows enhancement after intravenous (iv) contrast administration [29,34]. In the presence of liver cirrhosis or when a lesion measures less than 1 cm, the characterization can be challenging [35].

#### 6.3.4. Magnetic Resonance Imaging (MRI)

MRI is considered to be the method of choice in order to characterize all three types of CCA. All three types of CCA are hyperintense on a T2-weighted scan and hypointense on T1-weighted ones [31]. There is a variability of hyperintensity on T2-weighted images, which is due to the amount of fibrosis, mucin and necrosis within the tumor. These aspects depend on the subtype of the tumor: well/moderately/poorly differentiated adenocarcinoma. Occasionally, CCA can be isointense compared to the liver parenchyma on T1-weighted and T2-weighted images [36].

On diffusion-weighted imaging (DWI), CCA is peripherally hyperintense with a target-like appearance. This aspect is specific for CCA and is helpful in the differential diagnosis with HCC [36,37].

After iv contrast administration, the enhancement is similar to the one described on CT, including the enhancement’s variation depending on the degree of fibrosis [38].

An examination including gadolinium-enhanced MRI and magnetic resonance cholangiopancreatography (MRCP) is considered to be the optimal imaging examination when cholangiocarcinoma is suspected. It can provide information about biliary tract and liver anatomy, local extension, expansion of the biliary ducts, the vascular and lymph node involvement, liver metastases or contiguous organ invasion. MRCP has also the advantage of the 3D reconstruction of the bile ducts that offers a better characterization [36].

Intrahepatic CCA presents as a liver mass, measuring from 1 to 14 cm in diameter and has irregular or smooth margins and usually does not cause billiary obstruction. Perihilar and distal CCAs have an infiltrating periductal development pattern that leads to an irregular thickening of the bile duct wall, causing a narrowing at this level associated with the upstream dilatation of the intrahepatic bile ducts. At this point, it is important to differentiate the periductal infiltrating type from the intraductal lesions [36,39].

CCA is a hypovascular tumor and after the iv administration of gadolinium-based contrast agent, the enhancement pattern of iCCA is heterogeneous/homogeneous with prolonged and progressive delayed enhancement. In the delayed phase (5 min after iv gadolinum administration), the tumor can become isointense/mildly hyperintense compared to the adjacent liver parenchyma [36,40]. Due to the fact that CCA is a hypovascular tumor, washout is rarely seen after contrast administration. Initial enhancement and late phase peripheral washout are corresponding to regions of tumor cells, representing hypervascularity and increased perfusion but these aspects are rarely observed [41].

#### 6.3.5. Fluorodeoxyglucose Positron Emission Tomography (PET CT)

PET CT scans are useful for diagnosing CCA because the bile duct epithelium uptakes glucose but when small/periductal infiltrating tumors are present, the examination has limited utility. PET CT has a sensitivity of 60% and 90% for extrahepatic and intrahepatic CCAs, respectively, but has the greatest contribution in the detection of distant metastases, with a detection rate up to 100% [42].

### 6.4. Endoscopic Diagnostic Methods

#### 6.4.1. Endoscopic Ultrasound (EUS)

EUS is useful in order to detect a tumor, even when it is not detectable on CT or MRI. EUS has a higher sensitivity in detecting dCCA (81%) than pCCA (59%). Usually, the presence of a lesion extending through the bile duct wall, periductal infiltration or intraductal lesion and a bile duct wall thickness over 3 mm suggest the existence of CCA [43]. EUS is also able to provide T-staging with an accuracy ranging between 60% and 81% [44], N-staging with an accuracy ranging between 66% and 81% [45] and can detect major vascular invasion (100% accuracy) and the invasion into the adjacent organs [43]. Contrast-enhanced harmonic endoscopic ultrasonography (CH-EUS) improves the accuracy of T-staging. The behavior of CCA after contrast administration is the same with the one found at CEUS, showing hyperenhancement followed by a rapid washout. CH-EUS is not recommended for N-staging and M-staging due to the rapid washout [46]. Endoscopic Ultrasound-Guided Fine-Needle Aspiration (EUS-FNA) of the primary lesion or lymph nodes is useful for the tissue acquisition in order to establish the diagnosis. EUS-FNA of the primary lesion is discouraged due to the high risk of peritoneal tumor cell spreading [47]. A study with 16 patients that underwent EUS-FNA from the primary lesions concluded that 5 of 6 patients with a positive histological exam for adenocarcinoma had peritoneal metastases at the moment of surgery. EUS-FNA should be avoided especially when liver transplantation is an option of treatment [48].

Cytopathology has also an important role in the diagnosis of the pancreaticobiliary adenocarcinomas. In addition to the examination of conventional smears, the use of immunocytochemical markers is helpful. There are pancreaticobiliary lesions that are categorized as “atypical/suspicious and neoplastic”. A study conducted by Ieni et al. included 11 patients with lesions classified as indeterminate (with neoplastic cells found at the examination of smears) that could be correctly diagnosed by using cell-block preparation along with CD10/p53 immunostains. Immunocytochemistry against CD10 and p53 can be used in these cases with the purpose to distinguish gastrointestinal/pancreatic cellular contaminants that have hyperplastic or reactive alterations from differentiated pancreaticobiliary neoplastic elements [49].

Figure 1 and Figure 2 show two cases of EUS imaging of distal CCAs. Moreover, Figure 1b shows an elastography image with a blue-colored signal in the area of the lesion, suggesting the presence of hard tissue.

#### 6.4.2. Endoscopic Retrograde Cholangiopancreatography (ERCP)

ERCP is the most commonly used method for obtaining tissue confirmation of cholangiocarcinoma. The cholangiogram performed during ERCP shows a filling defect/stricture. The strictures are commonly asymmetric, irregular and long (≥10 mm). Brush cytology uses up to 10 passes and is considered to be safe; meanwhile, endobiliary biopsy usually uses 3 bites and offers larger tissue acquisition but has greater risks and can be technically challenging [50]. A meta-analysis comprising nine studies calculated the pooled sensitivity of brush cytology, intraductal biopsy, and both methods combined to be 45%, 48%, and 59%, respectively, with a specificity of 99%, 99%, and 100% [51].

Enhancing the cytological assessment of the collected piece is helpful in order to improve sensitivity. One method is by using FISH (fluorescence in situ hybridization) technology to analyze specific DNA sequences searching for known genetic aberrations. It can increase sensitivity from 20% to 43%, compared to routine cytology [52]. The triple assessment with brush cytology, endobiliary biopsy and FISH has a sensitivity of up to 82% [53].

Cholangioscopy has the advantage of direct visualization and biopsy of a stricture. This can be possible by using SpyGlass DS, which is a little endoscope (SpyScope) inserted through the working channel of the duodenoscope [53]. The biopsies collected using SpyGlass DS (Spybite) have a sensitivity of 64% [49]. This method is usually reserved for cases with indeterminate strictures with an inadequate initial biopsy collected through ERCP [54].

In case of indeterminate biliary strictures, the differential diagnosis between benign and neoplastic strictures can be made using the Monaco classification. This classification is using visual findings observed at cholangioscopy such as the presence of the stricture/lesion, the mucosa features, ulceration, papillary projections, abnormal vessels, pronounced pit and scarring. The malignant strictures usually have irregular margins, a dark mucosa with adherent mucous, the vessels are prominent and tortuous or interrupted and the pits are large, branched/disorganized, with dark centers. A study included 28 patients that performed cholangioscopy and 18 (64%) of them were diagnosed with malignant stricture (with positive histology). The patients diagnosed with malignant stricture had the Monaco criteria present at cholangioscopy as it follows: 39% had papillary projections and abnormal vessels; 18% had papillary projections, abnormal vessels and ulceration; 18% had only abnormal vessels; 14% had papillary projections and ulceration; 3.5% had abnormal vessels and ulceration; 3.5% had only papillary projections; and 3.5% of them had no lesion found during cholangioscopy. In conclusion, the Monaco criteria has a sensitivity of 66% and a specificity of 100% for malignancy, with a positive predictive value of 64% and a negative predictive value of 35% [55,56].

Table 1 summarizes the findings of 3 recent studies [57,58,59] on the diagnostic ability of EUS-FNA and ERCP tissue sampling in differentiating malignant from benign biliary strictures and one study [60] that evaluated the diagnostic ability of the two methods previously mentioned in differentiating extrahepatic CCA from benign biliary disease. These findings sustain the fact that EUS-FNA is superior to ERCP in differentiating malignant vs. benign biliary strictures, especially in extraductal lesions and lesions over 1.5 cm. When it comes to extrahepatic cholangiocarcinoma, ERCP tissue sampling (brush cytology or forceps biopsy) has superior specificity, PPV (positive predictive value), NPV (negative predictive value) and accuracy compared to EUS-FNA.

In conclusion, for diagnosing CCA, ERCP remains a better option than EUS-FNA. Moreover, it was proved that combining the two methods offers better sensibility, specificity, PPV, NPV and diagnostic accuracy.

#### 6.4.3. Confocal Laser Endomicroscopy

Probe-based confocal laser endomicroscopy (pCLE) can be possible by using a fiber-optic confocal laser in order to obtain higher magnification, showing epithelial cellular and subcellular structures, allowing a microscopic-level examination in real-time. This method has multiple applications in diagnosing gastrointestinal luminal neoplasia, including biliary neoplasia. The optical probe is used during ERCP or EUS and introduced into the biliary structures. It can provide images with a depth of view from 40 to 70 μm beneath the tissue surface and a field of view of 325 μm. Fluorescein is usually used as an intravenous contrast agent in pCLE in order to enhance the optical resolution of vascular arrangements and by quantifying the degree of contrast uptake in the biliary epithelium and subepithelium, the likelihood of malignancy can be determined in vivo [61].

#### 6.4.4. Intraductal Ultrasonography (IDUS)

Intraductal ultrasonography (IDUS) can assess the biliary tract and visualize the normal trilaminar structure. The disruption of the bile duct wall by a sessile mass that appears hypoechoic with irregular borders and lymphadenopathy suggests the likelihood of a malignant lesion [62]. A nodule greater than 8 mm, with or without an intact wall architecture, is suggestive of a malignant stricture. EUS is more accurate in diagnosing distal bile duct tumors and IDUS is more accurate in diagnosing proximal tumors [63,64]. Compared to ERCP, IDUS has higher sensitivity and specificity 87.5% and 90.6% vs. 62.5% and 53.1% [65]. Moreover, IDUS seems to have better sensitivity than EUS in T staging and predicting tumor resectability (89.1% and 81.8% vs. 75.6% and 75.6%; *p* < 0.002). Infiltrating cholangiocarcinomas are detected with a higher accuracy using IDUS-guided transpapillary forceps biopsy than biopsy through ERCP (detection rate 90.8% vs. 76.9%) [66].

## 7. Endoscopic Management

### 7.1. Endoscopic Guided Biliary Drainage

Biliary drainage in patients with jaundice can be achieved surgically by performing a biliary–enteric bypass or endoscopic/percutaneous stenting of the biliary tree. The placement of a stent through ERCP is preferred for both distal and more proximal biliary obstruction because it has similar rates of success and survival, but less morbidity compared to surgery [67].

Endoscopic guided biliary drainage is represented by endoscopic nasobiliary drainage (ENBD), endoscopic biliary stenting (EBS) and EUS-guided biliary drainage [68].

The majority of retrospective studies showed that there are no significant differences in decreasing jaundice and complication rate between percutaneous transhepatic biliary drainage (PTBD), ENBD and EBS [69].

#### 7.1.1. Endoscopic Biliary Stenting (EBS)

Until the 1980s, PTBD and surgical drainage were the main modalities for the palliation of obstructive jaundice in patients with cholangiocarcinoma. In 1980, endoscopic transpapillary biliary drainage was first introduced by Soehendra and Reynders-Frederix [70]. The first prospective, randomized comparison of PTBD and EBS using plastic stents was published by Speer et al. in 1987. This study proved that EBS had a significantly higher success rate (81% vs. 61%, *p* = 0.017) and a substantially lower 30-day mortality (15% vs. 33%, *p* = 0.016) [71].

In patients with CCA that causes the stricture of common bile duct, EBS can be used either as preoperative or palliative biliary drainage. Preoperative EBS is reserved for patients that have resectable tumors but need biliary decompression before the surgery either for acute cholangitis or to improve symptoms related to cholestasis (intense pruritus and jaundice) or if a delay of the surgery is expected (e.g., neoadjuvant therapy). According to ESGE (European Society of Gastrointestinal Endoscopy) recommendations, plastic stents, short intrapancreatic stents or covered self-expanding metal stents (SEMSs) may be used in this case. In patients who are candidates for neoadjuvant therapy, SEMSs are preferred. It was demonstrated that the insertion of a plastic stent followed by a delay in performing the surgery was associated with higher morbidity compared with surgery performed at 1 week after.

Palliative biliary drainage is used in patients with surgically incurable CCA and billiary obstruction. The aim of EBS in these cases is to cure acute cholangitis, to improve symptoms related to cholestasis and the medical status of patients prior to chemotherapy or radiotherapy.

In these cases, EBS is effective in over 80% of cases, with a lower morbidity compared to surgery. Compared to single plastic stents, SEMSs have a lower risk of recurrent biliary obstruction but there is no difference in patient survival. There is no difference clearly demonstrated between SEMS models with 10 mm diameter, even between covered and uncovered models. As for plastic stents, apparently the ones that measure 10 Fr in diameter, the DoubleLayer model and the plastic stents with an anti-reflux valve maintain the longest biliary patency. Moreover, it was demonstrated that no drug is recommended to lengthen the stent patency. According to ESGE recommendation, the drainage can be first attempted with a 10-Fr plastic stent if the diagnosis of malignancy is not established yet or if the patient has an expected survival of less than 4 months. Otherwise, if the diagnosis of malignancy was established, the initial insertion of a SEMS of 10 mm diameter is recommended if the expected survival is more than 4 months [72,73].

The Bismuth–Corlette classification is utilized for perihilar cholangiocarcinomas (CCAs), assessing the degree of ductal infiltration leading to strictures. It is also applied to evaluate benign strictures affecting the biliary ducts.

-Type I: the tumor is limited to the common hepatic duct below the confluence of the two hepatic ducts;-Type II: it involves the confluence of the two hepatic ducts;-Type IIIa: type II + involves the origin of the right hepatic duct;-Type IIIb: type II + involves the origin of the left hepatic duct;-Type IV: involves the origin of both hepatic ducts (type IIIa+IIIb)/multifocal involvement;-Type V: the stricture is localized at the junction of the common bile duct and the cystic duct [74].

If a malignant hilar stricture is present, the assessment of tumor resectability by MRI or CT should be performed in the absence of biliary stents because otherwise it can be affected. If the stricture is classified as Bismuth–Corlette type ≥ II, the biliary drainage can be achieved with fewer infective complications by PTBD compared to EBS. In high-volume centers that have multidisciplinary teams and experienced endoscopists, EBS should be performed instead of PTBD. In some cases, a combination of the two methods may be necessary in order to treat the infective complications of the strictures. It was demonstrated that draining over 50% of the liver volume shows a higher drainage effectiveness and also a longer survival than draining less than 50% of the volume. The liver sectors that will be drained should be selected based on MRI/CT before performing ERCP, the aim being the drainage of over 50% of the liver volume. The administration of antibiotics should be taken into consideration in case of anticipated or proven incomplete biliary drainage and continued until the achievement of complete drainage.

In patients with a malignant hilar stricture, the only recommended type of SEMS is the uncovered one, in order to prevent the occlusion of side branches. Uncovered SEMSs and plastic stents have similar short-term results in these cases, but SEMSs are preferred, providing a longer biliary patency. The choice of the stent depends on the decision about curative or palliative treatment. Plastic stents are recommended as long as no decision regarding the treatment has been taken. If the treatment will be palliative, uncovered SEMS are recommended if the life expectancy is over 3 months or if there is a biliary infection. In case of stent dysfunction, plastic stents are removed, the ducts are cleaned and after that, new stents are inserted. In case of dysfunction, the uncovered SEMSs are cleaned and if the stricture persists after that, new stents are inserted. For restenting, the choice between plastic stents or SEMSs will be based on the life expectancy and the degree of biliary infection [72]. The disadvantage of uncovered SEMSs is that they are not removable [75]. Regarding the malignant hilar biliary obstruction, a study conducted by Naitoh et al. has concluded that endoscopic bilateral drainage (using a SEMS) seems to be more effective than unilateral drainage (*p* = 0.009), but in terms of the placing technique of bilateral SEMS (stent-by-stent or stent-in-stent), there still are unsolved problems [76]. A randomized controlled trial has reported that the efficacy of bilateral stent-in-stent and stent-by-stent deployment appears to be similar regarding adverse events, clinical and technical success, stent patency and survival [77].

There are newer SEMSs that can be used for both hilar and distal malignant biliary strictures. Considering the fact that covered SEMSs have some limitations such as the risk of shortening after deployment and migration, a newer type, the laser-cut covered SEMS (LC-FCSEMS), has better deployment features, and is without the risk of shortening. An observational, retrospective, single-center study conducted by Marui et al. evaluated the efficacy of the newer LC-CSEMS (X-Suit NIR covered biliary metal stent by Olympus Medical Systems) in cases of unresectable malignant distal biliary stricture. These new SEMS are made of nitinol, have a 10 mm diameter and a 8–10 cm length, with an unique zigzag design (thick and thin struts). The thick ones maintain the stent form, providing a great radial force and maintaining the SEMS’ position and also the bile duct lumen. The role of the thin struts is to ensure the compliance of the SEMS, being able to fit into different bile ducts. The flared ends of the SEMS have the role to prevent migration. This new type has also a simple delivery system. This study included 52 patients, 75% of them had placement of a LC-CSEMS of 8 cm length. They had a follow up of 159 days and distal stent migration was observed in 2 patients, 6 patients had stent occlusion and 8 patients had recurrent biliary obstruction. Overall, this study had encouraging results of this new LC-CSEMS as an option for unresectable malignant distal biliary obstruction regarding the stent patency and reintervention. Additionally, LC-CSEMS can be very useful as well in hilar stricture because of its capacity of precise deployment. However, in this study, the LC-CSEMS used were not small-sized and their benefits have not been explored in hilar strictures, requiring further research [78].

Even if EBS is less invasive, there is the disadvantage of having a high incidence of ascending cholangitis, with a higher rate in patients with Bismuth type III and IV strictures [79].

Figure 3, Figure 4, Figure 5 and Figure 6 show different cases of biliary strictures in patients with CCA and different types of endoscopic biliary stenting.

#### 7.1.2. Endoscopic Nasobiliary Drainage (ENDB)

ENDB is a drainage method used in perihilar cholangiocarcinomas, along with EBS and PTBD. Even if PTBD has been extensively used with high success rates, the probable risk of seeding metastasis represents a serious problem. In most Japanese centers, ENBD is nowadays the first option for preoperative biliary drainage for perihilar CCA, according to the Japanese Clinical Practice Guidelines for Biliary Tract Cancer. ENBD causes nasopharyngeal irritation and that makes it uncomfortable for patients. That is the reason why ENBD is less used outside of Japan [80]. ENBD is superior in decreasing hyperbilirubinemia and is associated with a lower risk of catheter obstruction and ascending cholangitis, compared to endoscopic retrograde biliary drainage [81].

A study conducted by Kawashima et al. that involved 164 patients with perihilar CCA (including 128 patients with Bismuth type III and IV strictures) that had unilateral ENBD for preoperative biliary drainage. The technical and clinical success rates were 93.3% and 83.3%, cholangitis occurred in 28.8% of the cases and pancreatitis occurred in 20.1% of the cases [82].

#### 7.1.3. EUS-Guided Biliary Drainage (EUS-BD)

Even if ERCP is the therapeutic option of choice in non-surgical patients, EUS-BD can be an option when ERCP is not feasible [83]. EUS-guided biliary drainage (EUS-BD) has been used for unresectable CCA with a technical and clinical success rate of over 90% and over 70%, respectively [82]. EUS has an important role in the development of the endoscopic biliary drainage and represents an innovative and useful alternative method in the palliation of jaundice [84]. This can be possible by using lumen-apposing metal stents (LAMSs), which are self-expanding, fully covered devices that are able to form a stable anastomosis between neighboring organs and cavities. LAMS represents a new type of fully covered self-expandable metal stent which is mounted using a delivery system with an electrocautery distal tip, which facilitates the direct access to the biliary system [85,86]. The technique was first described in 2001 by Giovannini [83].

The main techniques of EUS-BD in malignant distal biliary obstruction are represented by cholecystogastrostomy, cholecystoduodenostomy and choledochoduodenostomy. Cholecystogastrostomy is indicated in patients with malignant distal biliary obstruction with retrograde dilatation of the common bile duct, usually when the gallblader is distented. It is also the simplest way of drainage because the gastric cavity offers more working space than the duodenum. Cholecystoduodenostomy has the same indications and efficiency as cholecystogastrostomy, only the technique is different. Choledochoduodenostomy is indicated when the common bile duct is distended as has a diameter of ≥15 mm. Valid for all the techniques of EUS-BD is that the distance between the target organ and the tip of the endoscope placed in the stomach/duodenum must be ≤10 mm. Also, after finding the optimal position a Doppler examination has to be performed in order to exclude the presence of vessels on the route where the LAMS will be inserted [83].

In perihilar CCA, in cases of surgically altered anatomy or if ulcers or strictures due to malignancy are present, an option of drainage using EUS can be hepaticogastrostomy. The aim is to create an anastomosis between the hepatic segment II or III and the stomach using a partially covered or a fully covered SEMS [87]. This technique can also be useful in complex hilar strictures with important invasion of the hilum and the right liver, in cases when ERCP is associated with a great risk of failure, as an initial drainage technique or as a rescue method after ERCP fails. Apparently, hepaticogastrostomy has shown comparative effectiveness to PTBD but has fewer adverse events and comparable success rate to ERCP [88,89]. The limitation of this method is that may not be effective in right-sided intrahepatic biliary obstruction or in complex hilar biliary strictures such as Bismuth type III or IV, because hepaticogastrostomy is a left-sided biliary decompression. The bridging method is combining ERCP and EUS, the guidewire is advanced using ERCP into the right hepatic bile duct. After the SEMS is placed between the right hepatic stricture and hepatic hilum, another metal stent will be placed from the left hepatic bile duct to the stomach (EUS-guided hepaticogastrostomy). In the locking stent method, an uncovered metal stent is placed between the right intrahepatic bile duct and the liver parenchyma, and a fully covered metal stent is inserted into the uncovered metal stent. After that, a fully covered metal stent is then placed between the proximal end of the uncovered metal stent and the duodenal bulb/stomach [90].

The main adverse events of hepaticogastrostomy are represented by abdominal pain, cholangitis, bile leakage, bleeding or self-limiting pneumoperitoneum, with an overall rate of 18%. Perforation and intraperitoneal migration of the stent are rarer than the above-mentioned adverse events [86]. As for the EUS-BD with LAMS techniques, the adverse events are represented by bleeding, migration, incorrect stent placement, perforation and infection, with an estimated overall rate of 16.3% [91].

Figure 7 is an EUS image showing EUS-guided deployment of the distal flange of a Hot AXIOS stent in the common bile duct, performing a choledochoduodenostomy in a case of distal CCA.

Figure 8a,b are EUS images showing the EUS-guided deployment of the distal flange of a Hot AXIOS stent in the cholecyst, performing a cholecystogastrostomy in a case of distal CCA.

### 7.2. Photodynamic Therapy (PDT)

PDT is an ablative method that consists in cellular apoptosis/necrosis in cells that are able to absorb a photosensitizer (absorbed preferentially by malignant/pre-malignant cells), which is activated by light of a specific wavelength. PDT in CCA requires the systemic administration of the photosensitizing agent followed 48–96 h later by ERCP/cholangioscopy with transpapillary placing of a laser-emitting diode into the bile duct. At the moment the diode is activated, it has the ability to emit a wavelength of 630 nm and when it is conducted towards the cells that have absorbed the photosensitizer, cellular apoptosis and necrosis occur [92].

A meta-analysis including ten studies had the purpose to evaluate the results of PDT combined with EBS compared to conventional EBS alone in palliation of unresectable CCA. The survival period in the PDT group was 413 days (95%CI: 349.54–476.54), which was superior to 183 days (95%CI: 136.81–230.02) in patients who underwent EBS alone [92].

One of the limitations of PDT is represented by the phototoxicity of the skin, caused by the photosensitizer administration, which occurs in 0% to 25% of the cases and has a duration of 4 to 6 weeks. The patients need to avoid light exposure after photosensitizer administration in order to prevent phototoxicity. Further limitations are the fact that PDT requires a two-stage approach and the adverse events that can occur after PDT, cholangitis and hepatic abscess [92,93].

### 7.3. Radiofrequency Ablation (RFA)

RFA is using a technology that through a catheter/probe delivers thermal energy to the malignant tissue, followed by locoregional coagulative necrosis. The thermal energy can be delivered via percutaneous (PRFA), intraoperative and endoscopic routes. In patients with unresectable CCA, endoscopic biliary RFA (ERFA) or PRFA can be used as a palliative measure. ERCP needs to be performed before ERFA in order to measure the stricture’s diameter and length. After that, the RFA catheter (a bipolar device) will be passed through the working channel of the duodenoscope and deliver thermal energy into the biliary tree [92,94].

A meta-analysis compared the survival and stent patency of combined ERFA and EBS vs. EBS alone. It was demonstrated that patients in the ERFA group had a pooled mean survival of 12.0 ± 0.9 months and patients of the EBS-only group had a mean survival time of 6.8 ± 0.3 months. Regarding the stent patency, in the ERFA with EBS group was 5.9 months compared to 3.6 months in the EBS only group (*p* < 0.001).

The same meta-analysis also compared the ERFA and PDT survival in patients with unresectable CCA. All patients underwent concomitant EBS via ERCP or PTBD. The pooled median survival in the ERFA group was 11.3 months and in the PDT group was 8.5 months (*p* = 0.02) [91].

The adverse events of ERFA are represented by cholecystitis, cholangitis, pancreatitis (more frequent), biliary tract perforation, partial liver infarction and fatal bleeding (in rare cases), with a careful postprocedural follow-up being necessary in these cases [93].

## 8. Surgical Treatment

Surgery is, at present, the only modality that can cure cholangiocarcinoma and should be agreed by a multidisciplinary tumor board specialized in malignant hepatobiliary pathology. Basic surgical principles have to be applied, hence R0 resection (no tumor at the margin) is the aim. In some cases, this cannot be possible, so there is a high incidence rate of R1 resections, especially in pCCA. It is also considered a standard of care (SoC) resecting the appropriate lymph nodes, but the optimal extent of lymphadenectomy can vary. The surgical management of iCCA respects the previously mentioned basic surgical principles with strong recommendation of routine lymphadenectomy up to the hepato-duodenal ligament.

In patients with pCCA, the diagnosis and evaluation of resectability according to the Bismuth Corlette classification can be determined only by using surgical exploration. Apparently about 15% of patients that undergo surgical intervention for presumed pCCA have an autoimmune cholangiopathy. It is important to perform radiological imaging before ERCP/PTC because as we mentioned before, the inserted biliary stents can influence the correct diagnosis and the assessment of the extension of CCA. The most common approach for pCCA is the extended right hemi-hepatectomy. The right portal vein embolization is used to promote hypertrophy of the future liver remnant (the liver segments II and III). The extended left resection is more complex, but it is considered that the liver remnant represented by the segments VI and VII is adequate. The liver segment I drains into the ductal bifurcation and has to be removed in any curative-intent surgery. A partial or complete resection of the liver segment IVb or V has to be accompanied by lymphadenectomy of the hepatoduodenal ligament. Vascular resections are possible, but vascular invasion has an unfavorable effect on prognosis. Lymphadenectomy should be performed during any radical surgical procedure for CCA. In locally unresectable pCCA, liver transplantation may be an option but with neoadjuvant chemotherapy before the liver transplantation. Nevertheless, liver transplantation it is not considered to be a SoC in pCCA. In patients with dCCA, during the surgical intervention, it is necessary to remove the pancreatic head via a partial duodeno-pancreatectomy (Whipple’s procedure), and the draining lymph nodes along with an extended bile duct resection up to the hilum [95].

## 9. Chemotherapy and Immunotherapy

Chemotherapy options in CCA are different according to the stage. In patients who are candidates for surgery, adjuvant chemotherapy with Capecitabine is recommended. In patients that need neoadjuvant chemotherapy before the surgical treatment, Cisplatin-Gemcitabine +/− Durvalumab are recommended. For patients with advanced or metastatic disease, the chemotherapy options are represented by Cisplatin-Gemcitabine +/− Durvalumab or FOLFOX (folinic acid, fluorouracil, and oxaliplatin) or 5-FU (5-Fluorouracil) +/− Irinotecan. After molecular profiling is performed, immunotherapy can be started and guided according to the findings:-IDH-1 mutation: Ivosidenib;-FGFR2 fusions: Pemigatinib, Infigratinib, Futibatinib;-BRAF mutation: Dabrafenib, Trametinib;-microsatellite instability (MSI-H/dMMR): Pembrolizumab;-HER2/neu overexpressions: Trastuzumab-Pertuzumab [95].

## 10. Conclusions

The positive diagnosis and staging of cholangiocarcinoma can be made by using imagistic and endoscopic methods or the two of them combined. Ultrasound, computed tomography imaging, magnetic resonance imaging and fluorodeoxyglucose positron emission tomography are the most used imagistic methods in the diagnosis and staging of cholangiocarcinoma. Regarding the endoscopic diagnostic methods, we can say that they are using both endoscopic and imagistic procedures, such as ERCP, EUS or IDUS. Those are the methods that can provide a sample for the histological examination, in order to establish the diagnosis.

The endoscopic management of cholangiocarcinoma is related firstly to biliary drainage. Biliary drainage can be achieved by using different endoscopic methods such as ERCP, EUS or ENBD. Choosing the optimal biliary drainage option depends on many factors mentioned in this review along with a demonstration of images of different strictures and the biliary drainage methods chosen by the team in our clinic.

Secondly, there are therapies like PDT and ERFA that can be delivered directly into the biliary tree using endoscopy, being able to reduce cholestasis and provide a superior survival time compared to biliary stenting alone in patients with unresectable CCA. PDT and ERFA are successfully used as an adjuvant therapy in order to improve the results of EBS. Lastly, surgery is the only modality that can cure CCA. Chemotherapy and immunotherapy in CCA should be chosen depending on the stage of CCA and the molecular profiling.

## Figures and Tables

**Figure 1 diagnostics-14-00490-f001:**
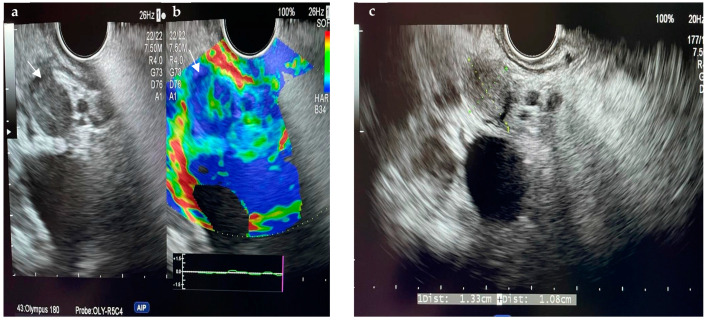
EUS imaging of a distal CCA, a lesion measuring 1.33 cm/1.08 cm marked by the white arrow (**a**,**c**), EUS+elastography showing a blue colored signal in the area of the lesion marked by the white arrow (**b**).

**Figure 2 diagnostics-14-00490-f002:**
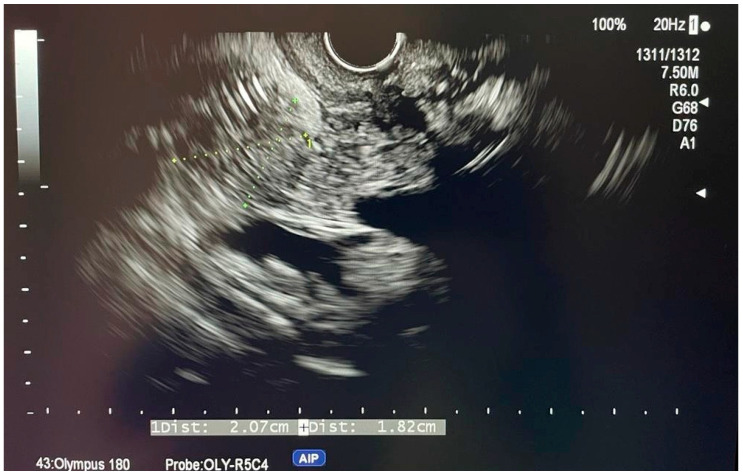
EUS imaging of a distal CCA, a lesion measuring 2.07 cm/1.82 cm and the dilated common bile duct.

**Figure 3 diagnostics-14-00490-f003:**
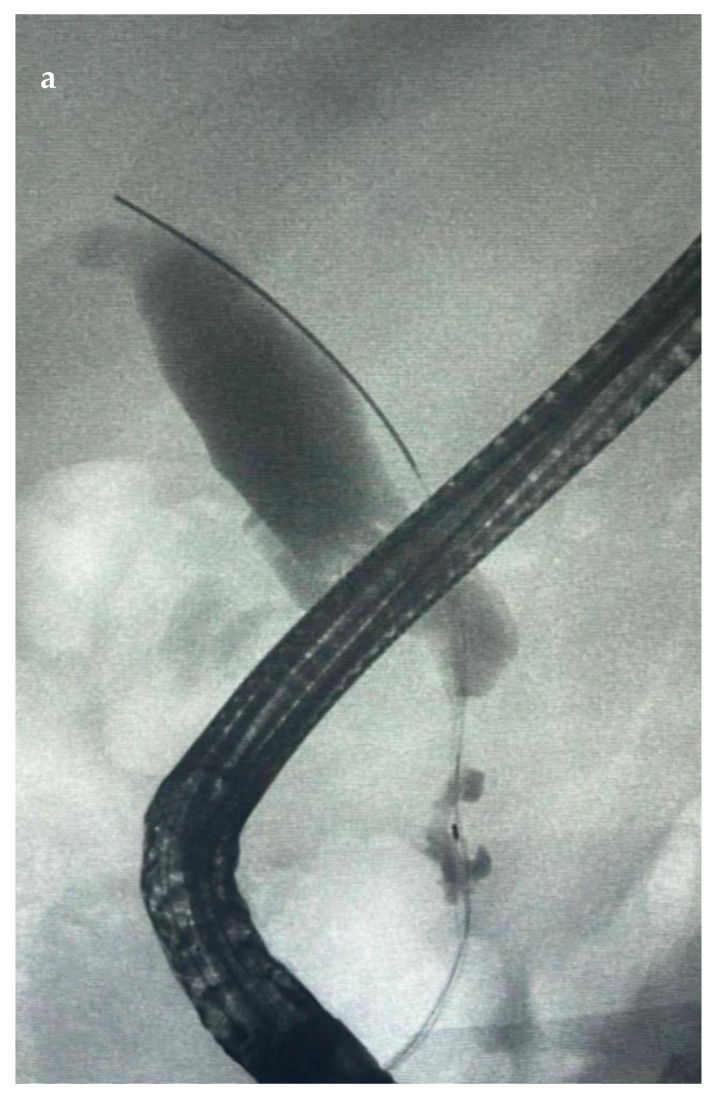
(**a**) Distal CCA with common bile stricture with (**b**) a fully covered SEMS used for drainage.

**Figure 4 diagnostics-14-00490-f004:**
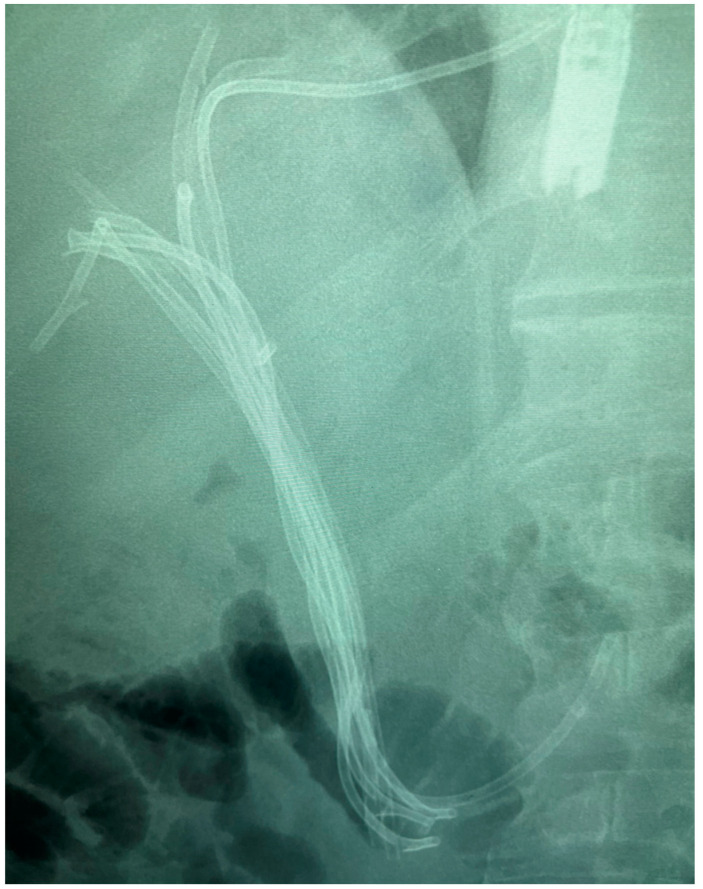
Perihilar CCA, Bismuth-Corlette type IV stricture with multiple plastic stents.

**Figure 5 diagnostics-14-00490-f005:**
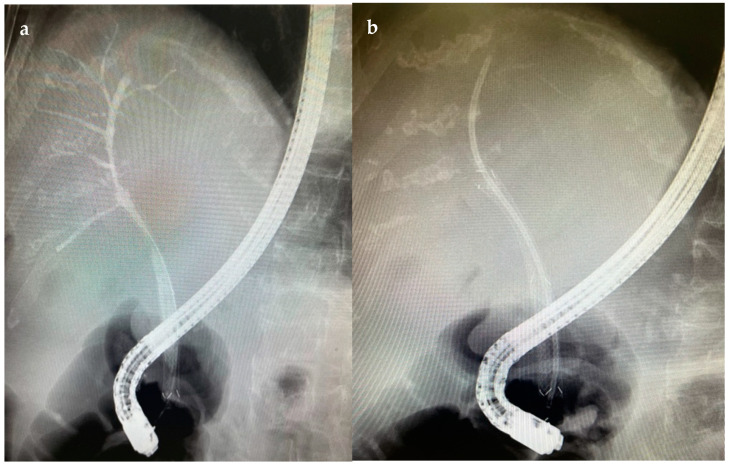
(**a**) Perihilar CCA, Bismuth–Corlette type IV stricture with uncovered SEMS invaded by the tumor; (**b**) 7 Fr/12 cm plastic stent placed inside the SEMS.

**Figure 6 diagnostics-14-00490-f006:**
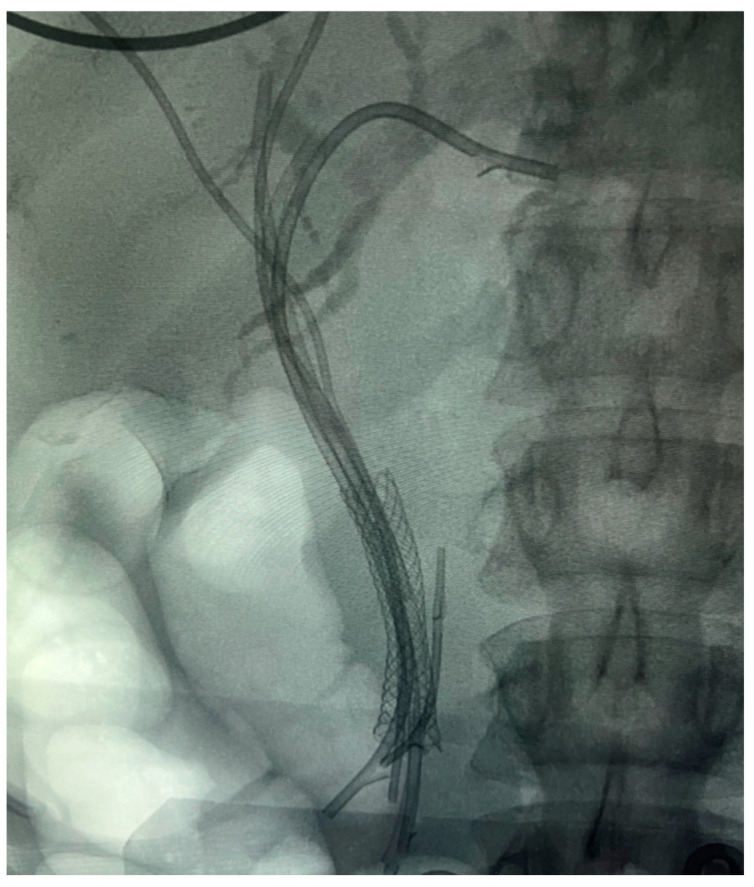
Intrahepatic CCA and obstruction of the common bile duct with multiple plastic stents and a fully covered SEMS 10/40 mm.

**Figure 7 diagnostics-14-00490-f007:**
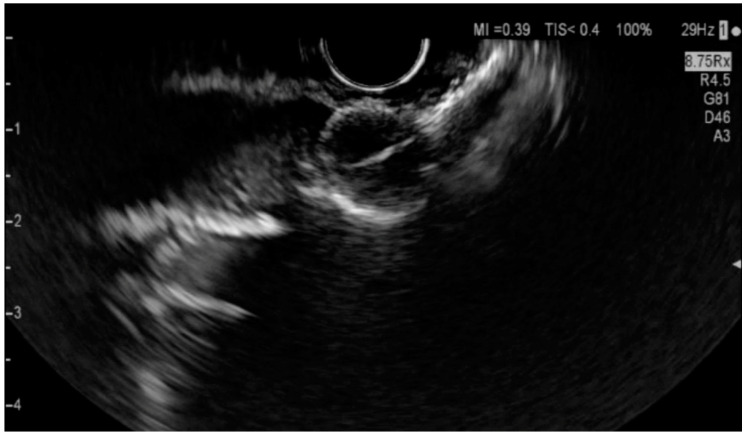
Choledochoduodenostomy. EUS-guided deployment of the Hot AXIOS stent in the common bile duct (distal flange).

**Figure 8 diagnostics-14-00490-f008:**
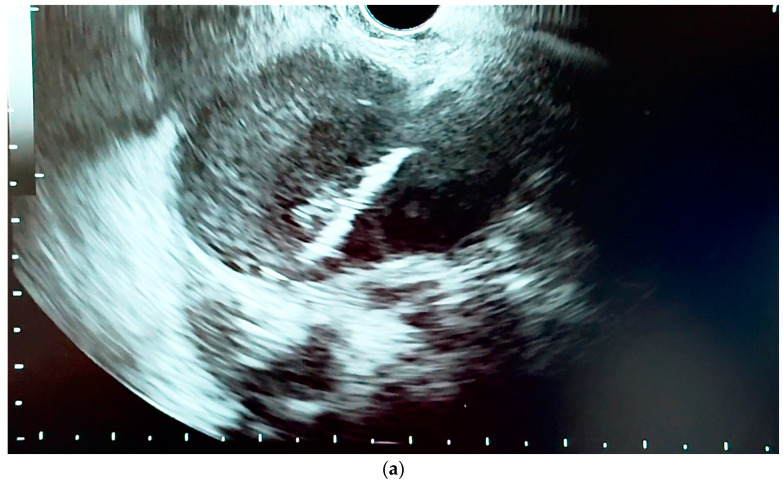
(**a,b**) Cholecystogastrostomy. EUS-guided deployment of the Hot AXIOS stent in the cholecyst (distal flange).

**Table 1 diagnostics-14-00490-t001:** Diagnostic ability of EUS-FNA and ERCP tissue sampling in differentiating malignant from benign biliary stricture and extrahepatic CCA from benign biliary disease.

Authors, Country (Year)	Method	Sensitivity, %	Specificity, %	PPV, %	NPV, %	Accuracy, %
Moura DTH., et al. Brazil (2018) [57]	ERCP tissue sampling (*n* = 61 patients)	60.4%	100%	100%	9.5%	62%
	EUS-FNA (*n* = 61 patients)	93.8%	100%	100%	40%	94%
Praveen M., et al. India (2022) [58]	ERCP brush cytology (*n* = 77 patients)	65.63%	100%	100%	37.14%	71.43%
	EUS-FNA (*n* = 77 patients)	90.63%	100%	100%	68.42%	92.21%
Sobhrakhshankhah E., et al., Iran (2021) [59]	ERCP brush cytology (*n* = 60 patients)	50.9%	100%	100%	15.6%	55%
	EUS-FNA (*n* = 60 patients)	78.2%	100%	100%	29.4%	80%
Onoyama T., et al. Japan (2019) [60]	ERCP tissue sampling (*n* = 54 patients)	76.0%	100%	100%	82.9%	88.9%
	EUS-FNA (*n* = 19 patients)	81.8%	87.5%	90%	77.8%	84.2%

## Data Availability

All data associated to this paper is contained within the article.

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
