# Peer review of "Updates in Diagnosis and Endoscopic Management of Cholangiocarcinoma"

_diagnostics, 2024, doi:10.3390/diagnostics14050490_

Round 1

Reviewer 1 Report

Comments and Suggestions for Authors

This is a comprehensive review on diagnostic and therapeutic role of endoscopy in the management of cholangiocarcinoma. The manuscript is well-written and presented.

Here are my comments to improve the manuscript. 

1) add some images of EUS findings in CCA

2) In the EUS-FNA section, add a table reporting the performance of available studies on this topic (e.g., PMIDs: 29876515, 36606017, 36407131, 30934706)

3) Also comment on the importance of sample processing by citing PMID 26063033 

4) In the ERCP section, add something about the visual classification of biliary lesions.

5) Add also something about new SEMS types for hilar strictures

Author Response

Firstly, I want to thank you for taking time to review our paper. 

1) I added 2 EUS images of distal cholangiocarcinomas and one image of elastography performed during EUS of one of the two cases (Figure 1 and 2)

2) I added the table reporting the performance of EUS-FNA compared to ERCP available in the studies suggested by you but I preferred to add it in the ERCP section, for the readers to have a better understanding of the two topics (Table 1)

3) I commented on the importance of sample processing by citing the article you mentioned. This part is found in text in the section 6.4.1. Endoscopic ultrasound (EUS) 

4) As for the visual classification of biliary lesions I added data about the Monaco criteria/classification that is using visual findings observed at cholangioscopy (found in text in the section 6.4.2. Endoscopic retrograde cholangiopancreatography (ERCP) )

5) I added data about the new laser-cut covered SEMS (LC-SEMS) that can be used in both distal and perihilar strictures (found in text in the section 7.1.1. Endoscopic biliary stenting (EBS) )

* I also added an EUS image and an image of direct endoscopic view of an EUS-guided choledochoduodenostomy in a case of distal cholangiocarcinoma

Reviewer 2 Report

Comments and Suggestions for Authors

A concise, excellent and comprehensive account of cholangiocarcinoma. All segments are presented, starting with epidemiological data, diagnosis and all endoscopic therapeutic modalities.

It is clear that the focus is on endoscopic treatment modalities, but it would be good to mention that chemotherapy and surgery also have their place in the treatment of CCA.

A few typos in the article need to be corrected.

My opinion is that the article provides clear guidelines for a problem-solving approach in everyday clinical practice and deserves publication.

Author Response

Firstly, I want to thank you for taking time to review our paper.

I added two new sections in the paper on the topic of surgical treatment and chemotherapy/immunotherapy options of CCA (section 8 and 9).

Thank you for suggesting these two topics because those were the missing topics that turn the article into a real mini guide for every gastroenterologist, as was its purpose.

Round 2

Reviewer 1 Report

Comments and Suggestions for Authors

I have no further comments